# Molecular Dynamics Simulation of the Synergistic Effect of Alkali/Surfactant/Polymer on the Formation and Stabilization of Water-Based Foam Systems

**DOI:** 10.3390/polym15030584

**Published:** 2023-01-23

**Authors:** Yong Wang, Xinpeng Le, Xingwang Wang, Wenbo Liu, Zhihua Wang

**Affiliations:** 1Key Laboratory for Enhanced Oil & Gas Recovery of the Ministry of Education, Northeast Petroleum University, Daqing 163318, China; 2College of Information Engineering, Shandong Vocational and Technical University of International Studies, Rizhao 276800, China; 3Petroleum Exploration and Production Research Institute, SINOPEC, Beijing 100083, China

**Keywords:** water-based foam, gas-liquid interface, molecular configuration, co-adsorption layer, liquid film drainage, molecular dynamics

## Abstract

The stable maintenance effect of a chemical oil displacement agent on a foam liquid film usually creates problems with the oilfields surface system. To achieve comprehensive insights into the influence mechanism of these chemical agent components on the foam liquid film, an “SDBS/HPAM/OH^−^” water-based foam simulation system and corresponding control systems were constructed by adjusting the categories and quantities of component molecules by molecular dynamics (MD) simulation. The simulated results indicated that the foam stability follows the order of “SDBS/HPAM/OH^−^” system > “SDBS/HPAM” system > “SDBS” system. The smaller the inclination angle of the SDBS molecular tail chain, the greater the tendency of the SDBS molecular configuration to be “upright” at the gas−liquid interface, which is not conducive to preventing the aggregation and penetration of gas molecules at the gas−liquid interface. Although the presence of HPAM molecules can significantly enhance the stability of the liquid film by restricting the liquid film’s drainage and the diffusion of gas molecules, the addition of HPAM molecules would weaken the formation ability of the foam liquid film. Through decreasing the aggregation of cations around the co-adsorption layer, OH^−^ not only enhances the interfacial activity of SDBS molecules, but also reduces the electrostatic repulsion between –COO^−^ groups on the HPAM molecular chain, which makes the foam more stable. With an increase in the pH, SDBS concentration, and HPAM concentration, the stability of foam liquid film was strengthened. These results are helpful in facilitating new insights into the formation and stabilization mechanism of water-based foams. In particular, they provide support for the development and application of new defoaming technologies.

## 1. Introduction

Foam is a multi-phase dispersion system in which many bubbles are separated by liquid. The necessary condition for its formation is the presence of both the foaming liquid phase and the foaming gas phase [1,2,3], as well as the introduction of external energy, including flow, agitation, shock oscillation and rapid depressurization [4]. The foam system can be classified according to the type of continuous phase medium. Foam which has a water phase as its continuous phase is a water-based foam, and the foam is oil-based when the continuous phase is an oil phase [5,6]. Foam is a typical thermodynamically unstable system; as soon as it is created, it is accompanied by evolution processes such as liquid drainage, bubble coalescence, and the rupture of the foam films. These processes are determined by the characteristics of the foam film and also affect the stability of the foam [7,8]. At present, the recognized foam decay and collapse mechanisms can be divided into foam drainage, gas diffusion, and foam coalescence. Although there are various factors affecting the stability of foam, they ultimately affect the stability of the foam liquid film through the mentioned forms [9,10]. Among them, foam drainage is the major factor leading to the foam instability. It can be attributed to the fact that the foam drainage reduces the liquid content of the foam liquid film, which leads to a decrease in liquid film thickness and the easier diffusion of gas through the liquid film. This is conducive to aggravating the degree of Ostwald ripening and causes the small foam to become larger; then, the large foam continues to expand and rupture [11,12,13]. Moreover, a change in foam size would further increase the drainage rate of the foam [14]. Compared with foam drainage and Ostwald ripening, the mechanism of foam coalescence is not well understood and recognized. However, it has been reported that the larger the size gap between adjacent bubbles, the more likely it is that the bubble coalescence will occur [15].

As a stable foam liquid film has a certain surface viscosity and adsorption ability, foam is widely used in materials, aerospace, medicine, cosmetics, construction, and other fields [16,17]. In the petrochemical industry, using the carrying characteristics of foam for oil, water, and sand, as well as some relatively mature applications including mineral flotation, foam flooding, foam plugging, and foam fracturing have been developed [18,19,20]. However, in the subsequent process of gathering, transportation, and treatment, the foam will blur the visible liquid level height in the separator, resulting in an increase in liquid level reading errors, which will further affect the measurement accuracy of the oil–water separator flowmeter [21]. Meanwhile, it will also create other problems such as gas corrosion and an increased working load in the separator, which seriously affect the treatment efficiency of the produced liquid and increase the production and operation costs of the oilfield [22,23,24]. In addition, with the development and application of chemical flooding technologies, the chemical agent components carried in the produced liquid—such as surfactants, including sodium dodecylbenzene sulfonate (SDBS); polymers, such as partially hydrolyzed polyacrylamide (HPAM), and alkali components—can decrease the free energy of the foam system and make the foam more stable [25,26,27]. Therefore, it is necessary to investigate the influence mechanism of these chemical agent components on the liquid film and to understand the formation and stabilization of the foam system so that the development and application of efficient defoaming technology can be developed.

Numerous studies have indicated that the foam formation and stability are closely related to the properties of the surfactants that are used to produce the foam [28,29]. Wang et al. [30] investigated the relationship between the gas flow rate and foaming ability and stability of an SDBS foam system. It was pointed out that, under a constant gas flow rate, when the SDBS concentration was less than the critical micelle concentration (CMC), the foaming ability and foam stability increased with the increasing SDBS concentration. When the SDBS concentration is higher than the CMC, increasing the SDBS concentration will no longer have an impact on the formation and stabilization of foam [30]. Polymers are the most common “foam stabilizers” used in combination with surfactants in foam systems. The polymers would be adsorbed on the liquid film surface to form a polymer–surfactant layer, which more strongly restricts the liquid film drainage and decreases the penetration of gas molecules into the gas-liquid interface, improving the stability of the foam system [31]. Experimental studies by Petkova et al. confirmed that the foaming ability and foam stability of a mixed solution of cationic surfactants and cationic polymers were enhanced [32]. However, there is no comprehensive report on the impact of the composition and properties of components including surfactants, polymers, and alkalis components on the foam liquid film, as well as their synergistic effects on the stability of the foam system. In particular, the effect of the pH variation of the system on the gas–liquid interface, and the interaction mechanism between the OH^-^ produced by alkali electrolysis and the surfactants and polymer, have not been systematically studied and recognized.

In this study, since oil displacement agent molecules are arranged and aggregated on the surface of a liquid film, the “SDBS/HPAM/OH^−^” water-based foam simulation systems and corresponding control systems were constructed by adjusting the categories and quantities of component molecules. The molecular dynamics (MD) simulation method was used to reveal the micromechanisms of different component categories, component properties, and their interactions that affected foam stability. The drainage behavior and Ostwald ripening process of the foam liquid film were described by means of characterization methods including the radial distribution function, diffusion coefficient, mean square displacement, interface formation energy, and molecular tail chain inclination angle. The stability of foam systems composed of different liquid phase components was compared, and the effect of the surfactant, polymer, and alkali agent on the foam liquid film were explained. The mechanisms behind component properties including pH, SDBS concentration, and HPAM concentration on foam liquid film’s stability were revealed. This is beneficial to the development and design of efficient oil-gas-water separation technologies for oilfield surface systems.

## 2. MD Simulation

### 2.1. Construction of the Simulation Systems

To simulate the gas phase of the water-based foam system of chemical flooding produced liquid, nitrogen, which is generally considered to have excellent foaming stability, was selected as the foaming gas, and the corresponding gas density was 1.145 g/L. For the liquid phase of the water-based foam system, the typical anionic surfactant sodium dodecylbenzene sulfonate (SDBS) (Petrochina Daqing Refining & Petrochemical Company, Daqing, China) was adopted as the foaming agent. Partially hydrolyzed polyacrylamide (HPAM) (Petrochina Daqing Refining & Petrochemical Company), with acrylamide (C_3_H_5_NO) (Xilong Chemicals, Guangzhou, China) and sodium acrylate (C_3_H_3_O_2_Na) (Xilong Chemicals) as the repeating units, was adopted as the polymer molecule in this work. Using the repeating units, HPAM molecules with a polymerization degree of 20 and hydrolysis degree of 20% were constructed by random polymerization. The construction of water molecules referred to the SPC model [33,34,35]. OH^−^ was introduced to simulate the alkaline environmental conditions of the foam system, and an equal amount of Na^+^ was added accordingly to ensure that the foam system was electrically neutral. Moreover, inorganic ions such as Ca^2+^, Mg^2+^, K^+^, Na^+^, Cl^−^, CO_3_^2−^, HCO_3_^−^, and SO_4_^2−^ were also constructed, referring to our previous work [36,37].

The construction process of the basic “SDBS/HPAM/OH^−^” multi-component foam simulation system is shown in Figure 1. Both to ensure the accuracy of the results and to save the simulation time, a suitable foam system size of 60 Å × 40 Å × 160 Å (*X × Y × Z*) was determined after repeated tests. The 48 surfactant molecules were vertically arranged at the gas-liquid interface according to the limit occupied area of molecule [38]. At that time, the interfacial adsorption concentration was 1.67 µmol/m^2^, and SDBS molecules were in the saturated adsorption state at the gas-liquid interface, reaching the CMC. Previous work confirmed that the foaming performance of the foam was the best at this time, making it suitable for the simulation research. Once the surfactant concentration exceeds the CMC, the SDBS molecular micelles are formed and their interfacial activity will then decrease significantly [39]. Two HPAM molecules were horizontally arranged on the gas-liquid surface. Then, 10 OH^−^ were filled into the water phase, and the corresponding amount of Na^+^ was added to ensure that the system was electrically neutral. Meanwhile, the water phase also contained 5 Ca^2+^, 5 Mg^2+^, 50 K^+^, 50 Na^+^, 100 Cl^−^, 32 HCO_3_^−^, 5 CO_3_^2−^, and 5 SO_4_^2−^.

Moreover, to explore the effects of the component composition and component properties on the stability of the water-based foam, different control simulation systems were set up based on the basic “SDBS/HPAM/OH^−^” foam simulation system by regulating the categories and quantities of component molecules. The details of the simulation schemes are shown in Table 1.

### 2.2. Simulation Details

Materials Studio 8.0 (Accelrys, San Diego, CA, USA) was chosen as the tool to perform all simulation programs [40,41]. The potential functions of all atoms and the interaction parameters between atoms in the system were allocated by the Compass II force field [42,43,44,45]. The periodic boundary condition was used in each direction of the simulation system. The velocity of each molecule in the initial state was randomly given according to the Boltzmann distribution. The long-range Coulomb interaction between particles adopted the Ewald algorithm. The atom-based algorithm was adopted for the Van der Waals interaction between particles with a cut-off radius of 1.55 nm. The smart Minimizer method was used to optimize the initial system in 20,000 steps to eliminate the local high-energy configuration, and it was annealed in five cycles from 300 K to 500 K, with a duration of 1000 ps to reduce the adverse effects of molecular overlap and intermolecular internal stress. Then, the Nose-Hoover thermostat algorithm was used to perform 1000 ps dynamic relaxation for the simulated systems in the NVT ensemble. During the process of simulation, the relaxation temperature was set to 298 K. The simulation step length and the time interval for trajectory collection were 1 fs and 1 ps, respectively.

## 3. Results and Discussion

### 3.1. Comparison of Liquid Film Stability of Different Foam Systems

#### 3.1.1. Interaction between Co-Adsorption Layer in Liquid Film and Water Molecules

Due to the polarity of water molecules in the foam film, when the SDBS molecules with a negative charge in the head groups are adsorbed and arranged on the surface of the foam film, a certain number of coordination water molecules would be attracted around the SDBS molecular head group to form a hydration shell caused by the hydration between the dipoles of water molecules and the SDBS head groups. The binding effect of the adsorption layer on water molecules in the hydration shell reduces its movement ability and diffusivity significantly, and slows down the liquid film drainage behavior, which promotes the sustainable existence of the foam liquid film [46,47,48]. The radial distribution function (RDF) is to describe the distribution probability of other particles around an object in a specified space. Therefore, the radial distribution function can be used both to reflect the order of matter and to reveal the interaction intensity between different particles. The stronger the interaction, the higher the peak value of the corresponding function [49].

Figure 2 shows the radial distribution function of water molecules around the SDBS head group in different foam systems. It can be seen that all foam systems form sharp peaks near position 2.2 Å, which indicates the formation of a hydration shell in the liquid film. Moreover, the RDF curve of the “SDBS/HPAM/OH^−^” system lies above that of other foam systems, and the peak value of its curve is the largest as well, indicating that the water molecules around the co-adsorption layer are aggregated to a high degree, and the water molecules within the liquid film are not easily lost. In contrast, the RDF curve of the “SDBS” system fell below that of other foam systems, and the peak value of the RDF curve was the smallest, indicating that the aggregation degree of water molecules around the co-adsorption layer in the liquid film decreased, and the attraction and binding ability of foam liquid film to water molecules decreased. Water molecules were easily lost through evaporation, drainage, and other forms, and foam stability was weakened.

Mean square displacement (MSD) reflects the degree of deviation between the spatial position of a particle and its initial position in the molecular dynamic simulation, which also can characterize the movement and diffusion properties of particles in the microscopic system, so it is another key parameter to describe the liquid drainage behavior of foam film. The larger the slope of the MSD curve, the stronger the particle movement and diffusion ability, the faster the liquid film drainage rate, and the worse the liquid film stability of the foam system [20]. The MSD can be written as Equation (1) [50]. Moreover, the MSD curves can be converted to the diffusion coefficients *D* of the corresponding particles by the Stokes-Einstein formula to achieve the quantitative characterization of the water molecules’ motion properties within the liquid film, and the conversion formula is as shown in Equation (2) [51].
(1)$$\mathrm{MSD}(t)=\frac{1}{N}\langle {\sum_{i=1}^{N}\left|r_{i}(t)-r_{i}(0)\right|}^{2}\rangle$$
(2)$$D=\frac{1}{6N}\underset{t\to \infty }{\lim}\frac{d}{dt}\sum_{N}^{i=1}{\left[r_{i}(t)-r_{i}(0)\right]}^{2}$$
where *N* stands for the number of diffusing molecules in the simulation system; *r_i_*(*t*) is the position of particle *i* at time *t*; *r_i_*(0) is the position of particle *i* at the initial stage.

Figure 3 shows the MSD curve of water molecules in the hydration shell of the liquid film in different foam systems, and the corresponding diffusion coefficients *D* are shown in Table 2. It can be seen that the MSD curve slope of water molecules in the “SDBS/HPAM/OH^−^” system is the lowest, and the diffusion coefficient is the lowest, which is 1.86 × 10^−5^ cm^2^·s^−1^, which indicates that the movement and diffusion rate of water molecules in the hydration shell in the foam liquid film is slow, the film drainage rate is low, and the foam stability is strong. In contrast, the slope of the MSD curve of the “SDBS” system and the “SDBS/HPAM” system increases, and the diffusion coefficient increases to 2.06 × 10^−5^ cm^2^·s^−1^ and 1.94 × 10^−5^ cm^2^·s^−1^, respectively. This indicates that the binding ability of the liquid film’s co-adsorption layer to the water molecules in the hydration shell is decreased, and the movement ability of water molecules in the liquid film is enhanced, which will accelerate the film drainage rate and inevitably decrease the stability of the liquid film of the foam system.

#### 3.1.2. Interaction between Co-Adsorption Layer in Liquid Film and N_2_ Molecules

In the snapshots of the different foam systems after the relaxation equilibrium as shown in Figure 4, the distribution characteristics of the nitrogen molecules can be observed. Although SDBS molecules are orientally adsorbed on the surface of the liquid film in the form of a head group extending into the water phase, forming an SDBS layer with an average thickness of 22.323 Å, this cannot effectively prevent the penetration of N_2_ molecules. As a result, a large number of N_2_ molecules diffuse in the SDBS layer, and some of them even penetrate into the water phase layer. This penetration can promote Ostwald ripening of the foam system and therefore reduce the foam stability. Figure 5 shows the radial distribution function of N_2_ molecules around water molecules in the liquid film of different foam systems. It can be used to characterize the aggregation and distribution characteristics of N_2_ molecules at the gas-liquid interface, and further indicate the degree of penetration of N_2_ molecules into the liquid films of different foam systems. It can be seen that the RDF function curves of the three systems have similar variation characteristics, namely a sharp rise in the curves from the 2 Å position. This indicates that the distribution characteristics of N_2_ molecules are similar in the liquid film, and that the N_2_ molecules diffuse and permeate the water phase up to a distance of 2 Å. Moreover, the RDF curve of the “SDBS” system lies slightly above that of the other foam systems, indicating that the diffusion degree of N_2_ molecules in the liquid film is higher than that in the “SDBS/HPAM” system and “SDBS-HPAM/OH^−^” system. This can be attributed to the hindering effect that the HPAM molecules on the surface of the liquid film have on the diffusion of N_2_ molecules, and consequently to their effect on the probability of instability caused by the Ostwald ripening of the ”SDBS” system, which is higher than that of other foam systems. This is also consistent with the aforementioned findings regarding the liquid film drainage characteristics. In summary, combined with the aforementioned diffusion characteristics of water molecules in the liquid film, it can be concluded that the foam stability of the “SDBS/HPAM/OH^−^” system is the highest, followed by the “SDBS/HPAM” system and the “SDBS” system.

### 3.2. Effect of Component Composition on the Stabilization of Water-Based Foams

#### 3.2.1. SDBS Molecules

Due to the presence of adsorption layers composed of SDBS molecules in the liquid films of both the “SDBS” system and “SDBS/HPAM” system, the adsorption configuration and spatial orientation of SDBS molecules at the gas-liquid interface can be characterized by the statistical tail chain inclination angle distribution [48,52,53,54].

The distribution of the SDBS molecular tail chain inclination angles on the co-adsorption layer of the different component foam systems is shown in Figure 6. It can be seen that the distribution range of the SDBS molecular tail chain inclination angle in the “SDBS” system is 56° to 60° which is smaller than in the other systems. This indicates that the SDBS molecules on the liquid film surface tend to be more “upright” at the gas–liquid interface, which is not conducive to preventing the aggregation and penetration of N_2_ molecules at the gas–liquid interface. The distribution ranges of the SDBS molecular tail chain inclination angle in the “SDBS/HPAM” system and “SDBS/HPAM/OH^−^” system are from 58° to 61° and 58° to 63°, respectively. The increase in the inclination angle indicates that the adsorption configuration of SDBS molecules on the liquid film tends to be flattened on the gas-liquid interface. The overlapping and winding molecular tail chains contribute to a denser protective layer at the gas-liquid interface, which further prevents the aggregation and penetration of gas molecules at the gas–liquid interface and decreases the possibility of Ostwald ripening in the foam film. This is one of the reasons that the aggregation degree of N_2_ molecules around the liquid film is lower in the “SDBS/HPAM” and “SDBS/HPAM/OH^−^” systems than it is in the “SDBS” system.

#### 3.2.2. HPAM Molecules

Figure 7 shows the instantaneous snapshots of the SDBS molecules in the adsorption monolayer in the “SDBS” system as well as the SDBS and HPAM molecules in the co-adsorption layer in the “SDBS/HPAM” system at the gas–liquid interface after dynamics reaching the relaxation equilibrium. Unlike the many “voids” appearing in the SDBS adsorption monolayer on the liquid film of the “SDBS” system in Figure 7a, the “voids” on the liquid film of the “SDBS/HPAM” system are filled with the stretched folding HPAM molecular chain adsorbed on the interfacial film, as shown in Figure 7b. The voidage of the liquid film surface of the foam decreases significantly, which leads to the formation of a denser and less permeable protective layer on the liquid film surface of the “SDBS/HPAM” system, which inevitably decreases the aggregation degree of N_2_ molecules at the gas–liquid interface of the foam film, and the possibility of N_2_ molecules penetrating the film and of the Ostwald ripening of the foam film is decreased as well. This is consistent with the comparison results of the distribution characteristics of N_2_ molecules in the different foam systems, and it also accounts for the finding that the aggregation degree of N_2_ molecules around the liquid film of the “SDBS/HPAM” system is lower than that of the “SDBS” system. Furthermore, as shown in Figure 8, the presence of N and O atoms in the HPAM molecule means that various types of hydrogen bonds are formed between the HPAM molecules and the surrounding water molecules. This intermolecular interaction leads to enhanced binding of the HPAM molecules to the water molecules, resulting in a decrease in the liquid film drainage rate and thus an increase in the foam stability.

Adding a surfactant into the interfacial system will decrease the interfacial tension and the total potential energy of the interface, which is conducive to the formation of a foam liquid film. Therefore, the higher the absolute value of the interface formation energy, the stronger the ability of surfactant molecules to decrease the energy of the gas–liquid interface, and the more beneficial it is to the formation of a liquid film [55,56,57]. Figure 9 shows the liquid film interface formation energy in various foam systems. It can be seen that the absolute value of the interface formation energy in the “SDBS/HPAM” system and the “SDBS/HPAM/OH^−^” system is lower than that of the “SDBS” system. The absolute value of the interface formation energy decreases from 566.50 kJ/mol in the “SDBS” system to 528.43 kJ/mol in the “SDBS/HPAM” system and 545.51 kJ/mol in the “SDBS/HPAM/OH^−^” system. This indicates that the generation ability of foam liquid film of the “SDBS/HPAM” system and the “SDBS/HPAM/OH^−^” system with the introduction of HPAM molecules decreases. Although the presence of HPAM molecules can significantly enhance the stability of the liquid film by restricting liquid film drainage and diffusion of N_2_ molecules, the addition of HPAM molecules would weaken the formation ability of the foam liquid film, which is also in agreement with the research findings and conclusions obtained from the previous work [58].

#### 3.2.3. OH^−^ Ions

Figure 10 shows the radial distribution function of OH^−^ with monovalent and divalent cations in the “SDBS/HPAM/OH^−^” system. It can be seen that the sharp peak formed by the RDF curve of OH^−^ and surrounding divalent cations in the liquid film of the foam system is higher than that of monovalent cations, and the first peak is located at 1.75 Å. Meanwhile, the first peak of the RDF curve of the OH^−^ surrounding monovalent cations is formed at 2.18 Å. This indicates that OH^−^ has a stronger aggregation degree with surrounding divalent cations and a closer interaction distance than monovalent cations. Combined with the MSD curves shown in Figure 11, the slopes of the MSD curves for OH^−^ and divalent cations are also closer, which both indicate the similar movement characteristics and diffusion behavior and the higher interaction strength of OH^−^ and divalent cations within the liquid film compared to the monovalent cations.

The instantaneous snapshots of the dynamic relaxation process of the “SDBS/HPAM/OH^−^” system are shown in Figure 12. For a clear view of the cation and co-adsorption layer, water and N_2_ molecules in these systems are concealed; the same is true in the following. As the simulation proceeds, divalent cations diffuse and move to the vicinity of the co-adsorbed layer, which is also attributed to the electrostatic interaction between the divalent cations and the molecules of the co-adsorbed layer. Meanwhile, some OH^−^ also diffuses to the co-adsorption layer of the gas–liquid interface. Furthermore, due to the strong electrostatic interaction between OH^−^ and divalent cations, it can be seen from the local detail of the snapshots that there is synergistic adsorption between the co-adsorbed layer molecules, OH^−^ ions and divalent cations when they are present simultaneously at the gas-liquid interface of the liquid film. Considering that OH^−^ and divalent cations can coexist in a solution in trace amounts and without forming precipitation, the existence of this synergistic adsorption will make it easier for divalent cations to enter the hydration shell and adsorb to the co-adsorption layer. On the one hand, it displaces some monovalent cations around the co-adsorption layer through competitive adsorption. On the other hand, divalent cations further decrease the aggregation degree of monovalent cations by occupying the spatial position of the ion layer, which leads to an increase in the interfacial activity of SDBS molecules, slows down the liquid film drainage behavior, and significantly enhances the liquid film generation ability and stability of the foam system. This is also consistent with the aforementioned results on the stabilization of the foam film and the distribution characteristics of monovalent cations around the adsorption layer.

Figure 13 shows the RDFs of the –COO^−^ of HPAM molecules and surrounding cations in the “SDBS/HPAM” system and “SDBS/HPAM/OH^−^” system. It can be seen that the peak of the RDF curve of the “SDBS/HPAM/OH^−^” system is lower than that of the “SDBS/HPAM” system, which indicates that the presence of OH^−^ reduces the aggregation degree of cations around the HPAM molecules in the co-adsorption layer and decreases the electrostatic shielding between the –COO^−^ groups on the HPAM molecules. The electrostatic repulsion between the –COO^−^ groups of the same molecular chain is enhanced, which causes the adsorption configuration of the HPAM molecules to be more stretched. This will contribute to an increase in the viscosity of the foam films, thus restricting the flow rate of water molecules, slowing down the liquid drainage behavior, and enhancing the foam stability. This is also in agreement with the results of the aforementioned comparison of the liquid film stability of the “SDBS/HPAM” system and “SDBS/HPAM/OH^−^” system.

### 3.3. Effect of Component Properties on the Stabilization of Water-Based Foams

#### 3.3.1. pH Values

Figure 14 shows the radial distribution function of the SDBS head group and surrounding water molecules in the liquid films of different pH foam systems. It can be seen that all RDF curves show a peak at position 1.47 Å, indicating the formation of hydration shell in the liquid film. Moreover, with the increase in pH, the peak value of the RDF curve increases. The RDF curve of the pH 10 foam system is above that of other foam systems. This indicates that the hydration shell has a high content of “bound water”, corresponding to the strong attraction and binding ability of water molecules by the interfacial film.

The coordination numbers of water molecules in the hydration shells of foam systems with different pH are shown in Table 3. It can be seen that the pH 7 foam system has the smallest water molecule coordination number of 1.61. With the increase in pH, the coordination number of water molecules increases from 1.74 in the pH 8 foam system to 1.80 in the pH 9 foam system and 1.82 in the pH 10 foam system, respectively. This indicates that the water molecules on the surface of the foam film are not easily lost through forms such as drainage and evaporation, and the co-adsorption layer has a strong ability to attract and bind the “bound water”, which is conducive to enhancing the foam stability. In summary, the higher the pH, the stronger the stability of the foam system.

When the pH increases, the number of OH^−^ in the interfacial film of the foam system increases, which inevitably leads to a change in cation aggregation and distribution around the co-adsorption layer, which then affects the overall stability of the liquid film. To further explain the influence mechanism of pH on foam stability, it is necessary to consider the fact that the monovalent cation is the main component of cations in the simulated water-based foam system. Therefore, as shown in Figure 15, the RDFs of monovalent cations in the liquid film with SDBS head groups in foam systems with different pH were established. It can be seen that the RDF curves all show a peak at position 1.97 Å, indicating that monovalent cations form an aggregated and adsorbed ion layer within the hydrated shell of the foam film. In addition, when the pH of the liquid phase of the foam system increases, the peak value of the RDF curve decreases, which indicates a decrease in the aggregation degree of the monovalent cations around the co-adsorption layer in the liquid films. This can be attributed to the increasing electrostatic attraction between OH^−^ and the cations in the liquid films caused by the number of free OH^−^ in the liquid film, which increases. As a result, the aggregation degree of cations around the co-adsorption layer decreases, and the ability of the foam liquid film co-adsorption layer to attract and bind “bound water” molecules is enhanced. Therefore, the higher the pH of the foam system, the more conducive it is to slowing the drainage behavior of the liquid film and enhancing the foam stability.

#### 3.3.2. SDBS Concentrations

Figure 16 shows the RDFs of the SDBS head group and surrounding water molecules in the liquid films of foam systems with different SDBS concentrations. It can be seen that all RDF curves show a peak at position 1.46 Å, indicating the formation of a hydration shell in the liquid film. Furthermore, the peak value of the RDF curves increases with the decrease in SDBS concentration. The RDF curve of the foam system with an SDBS concentration of 1.25 μmol/m^2^ is located above that of the other foam systems. The sharp peak value is 6.26, which indicates that there are more water molecules in the hydration shell around the co-adsorption layer in the liquid film. In contrast, the RDF curve of the foam system with an SDBS concentration of 1.67 μmol/m^2^ lies below that of the other foam systems, which reveals that there is a minimum quantity of “bound water” molecules around the co-adsorption layer in the liquid film, and that the water molecules are easily lost through drainage, evaporation, and other behaviors.

The coordination numbers of water molecules in the hydration shell in the liquid films of foam systems that have different SDBS concentrations are shown in Table 4. It can be seen that the coordination number of water molecules increases with the decrease in the SDBS interfacial adsorption concentration of the foam system. When the SDBS concentration in the foam system decreases from 1.67 μmol/m^2^ and 1.53 μmol/m^2^ to 1.39 μmol/m^2^ and 1.25 μmol/m^2^, the coordination number of water molecules in the hydrated shell increases from 1.74 and 1.79 to 1.93 and 1.98, respectively. This indicates that the diffusion ability of water molecules in the liquid film’s hydration shell is significantly enhanced, the film drainage rate becomes faster, and the foam stability decreases. This also means that the lower the SDBS concentration in the interfacial film, the weaker the stability of the foam system [13,36].

The instantaneous snapshots of the gas-liquid interface in foam systems with different SDBS concentrations after dynamics relaxation equilibrium are shown in Figure 17. It can be seen that the water molecules and N_2_ molecules at the gas-liquid interface are locally aggregated in the upper and lower liquid films. Through measuring the vertical distance of the water molecule aggregation area at the upper and lower gas–liquid interfaces along the *Z* axis, it can be seen that, when the SDBS concentration in the foam system decreases from 1.67 μmol/m^2^ and 1.53 μmol/m^2^ to 1.39 μmol/m^2^ and 1.25 μmol/m^2^, the vertical distance of the water molecule aggregation area at the upper and lower gas–liquid interfaces increases from 25.289 Å and 26.839 Å to 27.213 Å and 30.167 Å, respectively. When the interfacial adsorption concentration of SDBS in the foam system decreases to 1.25 μmol/m^2^, some water molecules even escape from the liquid phase and enter the gas phase. This indicates that, with the decrease in SDBS concentration, the binding ability of the protective layer formed by the foam liquid film co-adsorption layer to the water molecules in the liquid film decreases, and the diffusivity of water molecules increases. The water molecules at the gas-liquid interface tend to move away from the liquid phase to the gas phase, and then overflow the liquid film. This can be attributed to the decrease in the interfacial adsorption concentration of SDBS molecules, which also means that the number of SDBS molecules with hydration ability on the liquid film surface decreases, resulting in a significant decrease in the attraction and binding ability of the foam liquid film co-adsorption layer to water molecules. The lower the interfacial adsorption concentration of SDBS molecules in the interfacial film, the more active the water molecules at the gas-liquid interface, and the more obvious the tendency of the water molecules to overflow the interfacial film. Meanwhile, more water molecules with an overflow trend will increase the coordination number of water molecules around the co-adsorption layer when they diffuse into the gas phase through the SDBS head groups.

The sectional snapshots of the gas-liquid interface in foam systems with different SDBS concentrations are shown in Figure 18. It can be seen that the decrease in SDBS concentration also leads to a reduction in the area covered by SDBS molecules on the gas–liquid interface, and more voids appear on the liquid film surface. This not only makes it easy for some water molecules that are attracted at the surface of the liquid film and have weak binding strength to overflow into the gas phase through these voids, but also makes it easier for N_2_ molecules to penetrate the interfacial film. This increases the probability of Ostwald ripening and causing foam liquid film instability.

#### 3.3.3. HPAM Concentrations

Figure 19 shows the RDFs of the SDBS head group and surrounding water molecules of the liquid film in foam systems with different HPAM concentration. It can be seen that all RDF curves show a peak at position 2.13 Å, indicating the formation of a hydration shell in the liquid film. Moreover, the peak value of the RDF curve increases with the increasing HPAM concentration, which indicates an increase in the quantity of “bound water” molecules in the liquid film. The RDF curve of the “3-HPAM” foam system is located above that of other foam systems, and the sharp peak value is 5.76. The RDF curve of the “3-HPAM” foam system lies below that of the other foam systems, with the lowest peak value of 4.66, which indicates the lower content of “bound water” in the hydration shell of the liquid film. This also shows that the ability of the interfacial co-adsorption layer to attract and bind water molecules is decreased.

The coordination numbers of water molecules in the hydration shell in the liquid films of foam systems with different HPAM concentrations are shown in Table 5. As the HPAM concentration of the foam system increases, so do the coordination number of water molecules increases. The coordination number of water molecules in the hydration shell in the liquid film increases from 1.6 in the “0-HPAM” foam system and 1.63 for the “1-HPAM” foam system to 1.69 in the “2-HPAM” foam system and 1.86 in the “3-HPAM” foam system. This indicates that the number of “bound water” molecules around the co-adsorption layer in the corresponding liquid film increases, and the attraction and binding ability of the gas–liquid interface to the water molecules in the hydration shell is enhanced, so that the water molecules in the interfacial film are not easily lost due to various behaviors, which is beneficial to the foam’s stability. This also means that the higher the HPAM concentration in the foam liquid film, the stronger the foam stability.

The gyration radius can be used to characterize the size and bending degree of polymer molecules. The larger the gyration radius, the higher the stretching of the molecules. It can be expressed as follows [59,60]:(3)$$R_{g}={\left(\frac{\sum_{i}{\left\|r_{i}\right\|}^{2}m_{i}}{\sum_{i}m_{i}}\right)}^{1/2}$$
where $m_{i}$ is the mass at point *i*, and $\left\|r_{i}\right\|$ is the vector from point *i* to the mass center of the molecular chain.

The gyration radius of HPAM molecules in foam systems with different HPAM concentrations are shown in Table 6. It can be seen that, with the increase in the HPAM concentration in the foam liquid film, the gyration radius of HPAM molecules decreased from 11.92 Å in the “1-HPAM” foam system to 11.45 Å in the “2-HPAM” foam system and 11.37 Å in the “3-HPAM” foam system. This indicates that the adsorption configuration of HPAM molecules in the interfacial co-adsorption layer of the foam system becomes curved, which does not seem conducive to increasing the viscosity of the liquid film and restricting the movement of water molecules. However, as the adsorption concentration of HPAM molecules on the co-adsorption layer increases, the number of –COO^−^ with hydration on the HPAM molecular chain increases, which can assist SDBS molecules in attracting and binding more water molecules in the co-adsorption layer. This is the major factor affecting the liquid film drainage behavior of foam systems with different HPAM concentrations.

Figure 20 shows the sectional snapshots of the gas–liquid interface in foam systems with different HPAM concentrations. It can be seen that, as the concentration of HPAM molecules in the liquid film increases, the polymer molecular chain plays the role of synergistically filling the “voids” formed by the inhomogeneous distribution of co-adsorbed layer molecules on the interface. The specific finding is that the area of the surface of the liquid film is not covered by the decreased co-adsorption layer molecules. This inevitably decreases the voidage of the liquid film surface and hinders the diffusion and penetration of N_2_ molecules in the liquid film. At the same time, the increase in the number of HPAM molecules is beneficial to HPAM molecules, which use its flexible curling properties to hinder the flow of “bound water” at the gas-liquid interface and intercept the water molecules on the surface of the liquid film with the tendency to drain, restricting its movement rate, which further slows down the film’s drainage and maintains its stability. Obviously, there are various mechanisms that cause the increasing HPAM concentration to affect the restriction of the water molecule’s movement and the diffusion ability of the foam liquid film.

The distribution of SDBS molecular tail chain inclination angles in foam systems with different HPAM concentrations are shown in Figure 21. It can be seen that, when there is no HPAM molecule in the foam system, the tail chain inclination angle of SDBS molecules in the liquid film in the “0-HPAM” foam system is distributed around 32.49°. With the increase in the HPAM concentration in the foam system, the tail chain inclination angle of SDBS molecules increases gradually, i.e., the tail chain inclination angle distribution increases from 34.52° in the “1-HPAM” foam system to 34.53° in the “2-HPAM” foam system and 36.47° in the “3-HPAM” foam system. This indicates that, with the increasing HPAM concentration in the foam system, the adsorption configuration of SDBS molecules tends to spread on the gas–liquid interface. The change in the molecular configuration characteristics promotes the degree of mutual entanglement between the hydrophobic tail chains of SDBS molecules and HPAM molecules in the co-adsorption layer. The permeability of the co-adsorption layer on the liquid film surface decreases, and the N_2_ molecules are not easily diffused into the liquid film, thus enhancing the stability of the foam system. This can be attributed to the fact that both HPAM and SDBS contain long carbon chains in their molecular structures. The similar structure contributes to the tendency of the hydrophobic tail chains of SDBS molecules to approach the molecular chains of HAPM molecules when the concentration of HPAM in the co-adsorption layer increases, which leads to its molecular configuration being spread across the liquid film, thus enhancing the foam’s stability.

## 4. Conclusions

A comprehensive understanding of the impact of chemical oil flooding agents on the formation and stabilization of foam systems to facilitate the development and application of efficient defoaming technologies is key to improving the efficiency of oil-gas-water separation in oilfield surface systems. Therefore, in this paper, the micromechanisms of component categories, component properties, and their interactions in the liquid phase on the stability of foam liquid films was revealed by the molecular dynamic simulation method.

(1)The liquid film co-adsorption layer of the “SDBS/HPAM/OH^−^” system causes it to have a stronger ability to bind water molecules, which restricts the foam drainage rate, meaning that the foam liquid film has the strongest stability. The diffusion degree of N_2_ molecules in the liquid film of the “SDBS” system is higher than in other foam systems, and consequently the probability of instability caused by the Ostwald ripening of the “SDBS” system is higher than that in other foam systems.(2)Compared with the “SDBS” system and “SDBS/HPAM” system, the inclination angle of the SDBS molecular tail chain in the “SDBS/HPAM/OH^−^” system is larger, and the adsorption configuration of SDBS molecules on the interfacial film tends to be flattened on the gas-liquid interface, which makes the protective layer of the gas-liquid interface more dense. HPAM molecules play a role in foam stabilization by expanding the coverage area of the foam liquid film co-adsorption layer and cooperating with SDBS molecules to spread at the gas-liquid interface to restrict the N_2_ molecules permeability in the interfacial film. Through decreasing the aggregation of cations around the co-adsorption layer, OH^−^ not only enhances the interfacial activity of SDBS molecules, but also reduces the electrostatic repulsion between –COO^−^ groups on the HPAM molecules, which makes the adsorption configuration of HPAM molecules more stretched and increases the viscosity of the foam liquid film. These behaviors are favorable to enhancing the foam stability.(3)When the pH increases, the aggregation degree of monovalent cations around the co-adsorption layer of the foam liquid film decreases, and the interfacial activity of SDBS molecules in the co-adsorption layer increases. The ability to attract and bind the water molecules increases as well, which is beneficial to the alleviation of the foam liquid film’s drainage. As the concentration of SDBS decreases, the number of SDBS molecules adsorbed on the co-adsorption layer of the foam film decreases. At the same time, some “voids” are formed on the liquid film because its coverage area on the film surface decreases, which promotes the penetration of N_2_ molecules and weakens the formation ability and stability of the liquid film. With the increase in the HPAM concentration, the moving rate of water molecules in the gas-liquid interface of the liquid film decreases, the drainage of the interfacial film slows down, and the foam stability increases.

## Figures and Tables

**Figure 1 polymers-15-00584-f001:**
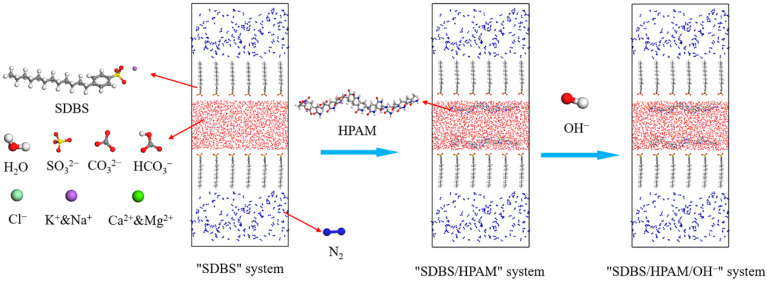
Illustration of simulation system construction.

**Figure 2 polymers-15-00584-f002:**
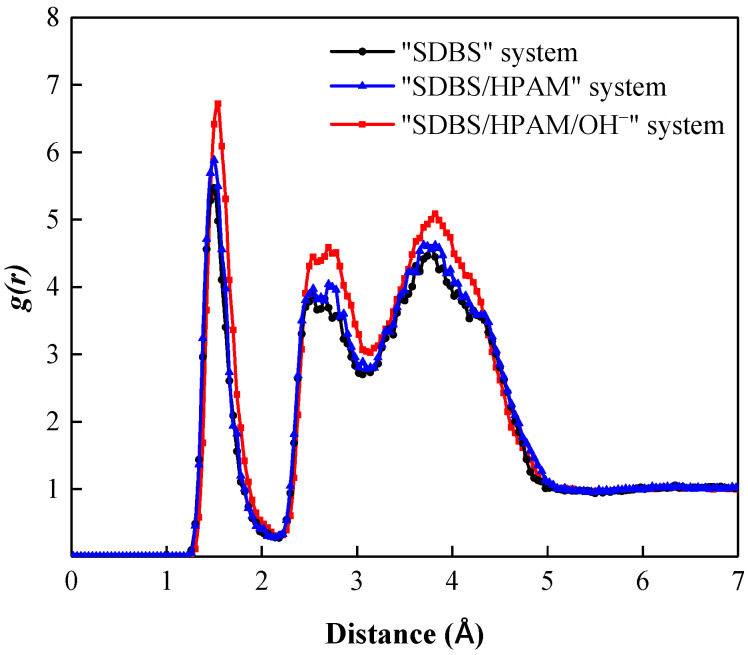
Radial distribution function of water molecules around SDBS head group in different foam systems.

**Figure 3 polymers-15-00584-f003:**
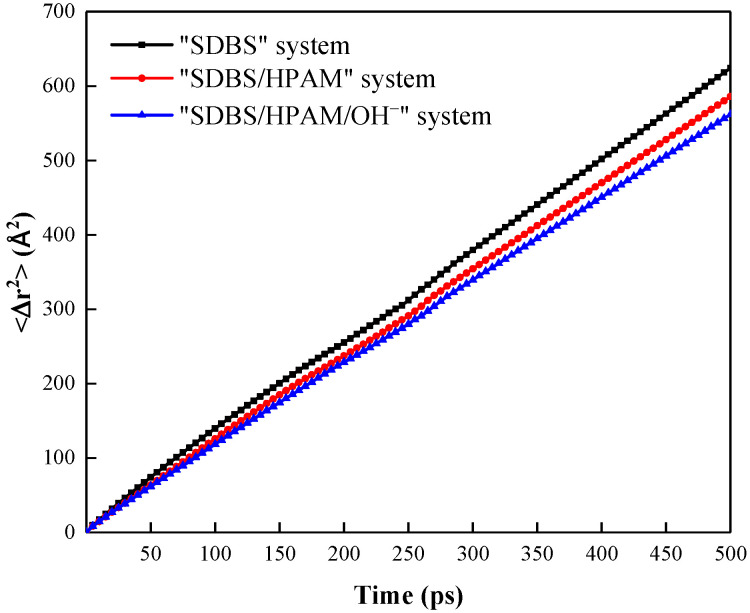
Mean square displacement of water molecules in the hydration shell of the liquid film in different foam systems.

**Figure 4 polymers-15-00584-f004:**
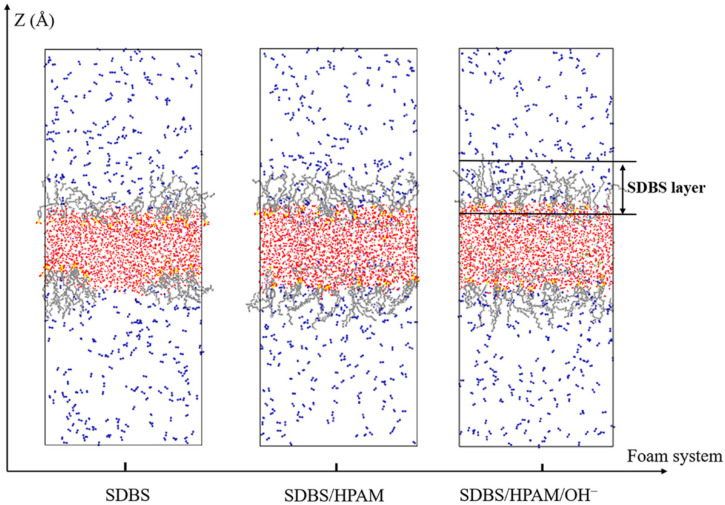
Snapshots of the different foam systems after the relaxation equilibrium.

**Figure 5 polymers-15-00584-f005:**
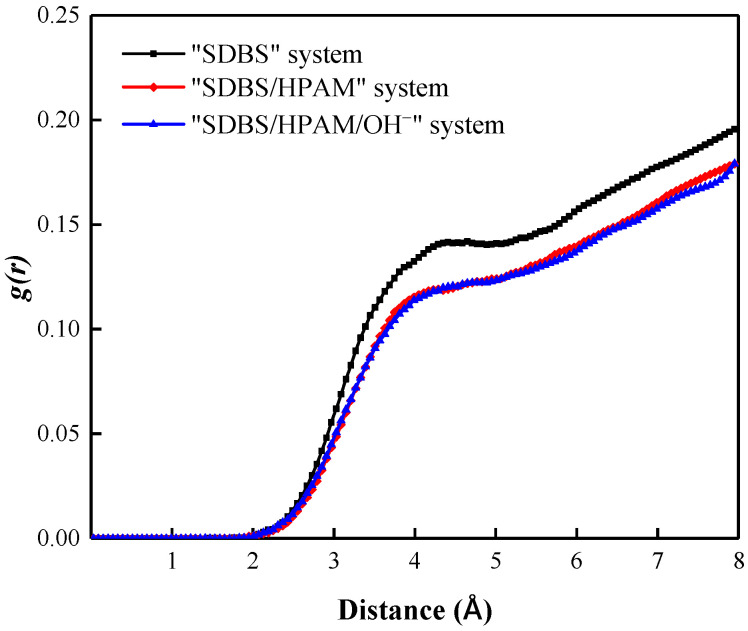
Radial distribution function of N_2_ molecules around water molecules of liquid films in different foam systems.

**Figure 6 polymers-15-00584-f006:**
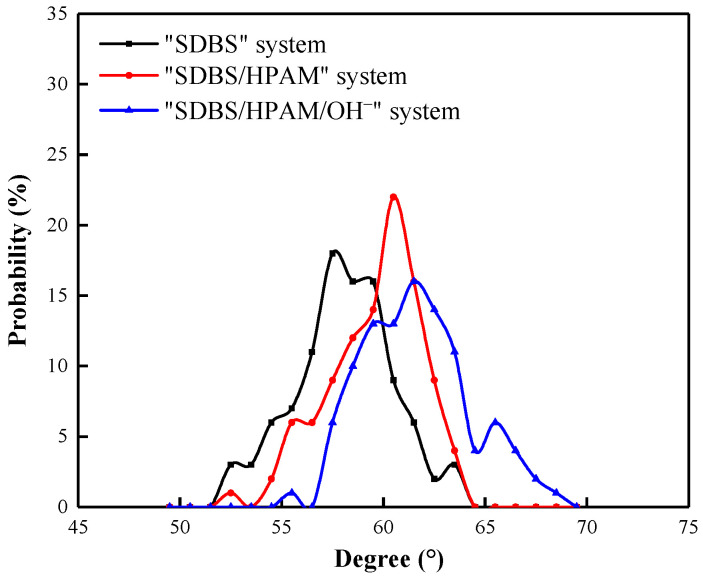
Distribution of SDBS molecular tail chain inclination angles in different component foam systems.

**Figure 7 polymers-15-00584-f007:**
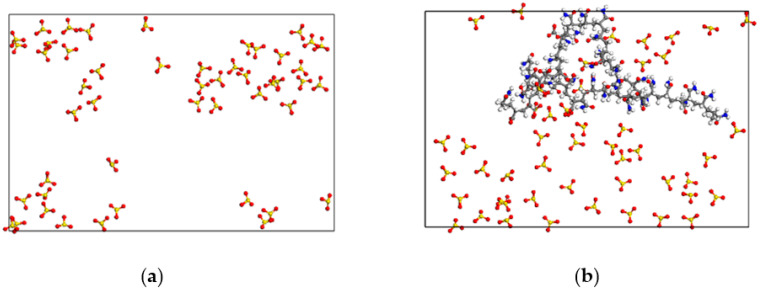
Snapshots of the molecular configuration of adsorption layer after dynamics relaxation equilibrium. (**a**) “SDBS” system; (**b**) “SDBS/HPAM” system.

**Figure 8 polymers-15-00584-f008:**
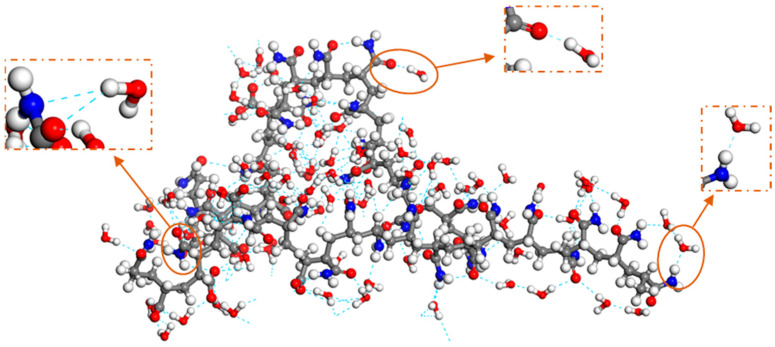
Snapshots of the formation hydrogen bond between HPAM and water molecules.

**Figure 9 polymers-15-00584-f009:**
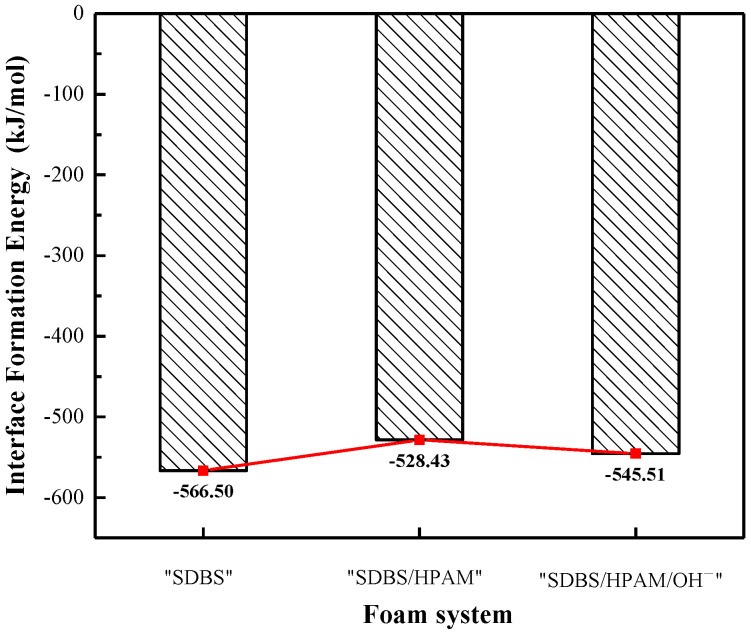
Interface formation energy in different foam systems.

**Figure 10 polymers-15-00584-f010:**
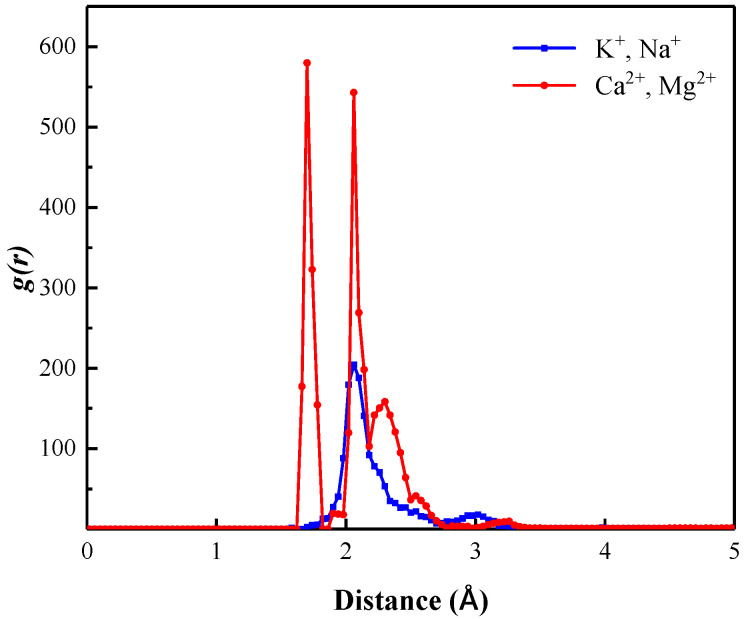
Radial distribution function of OH^−^ with monovalent and divalent cations in “SDBS/HPAM/OH^−^” system.

**Figure 11 polymers-15-00584-f011:**
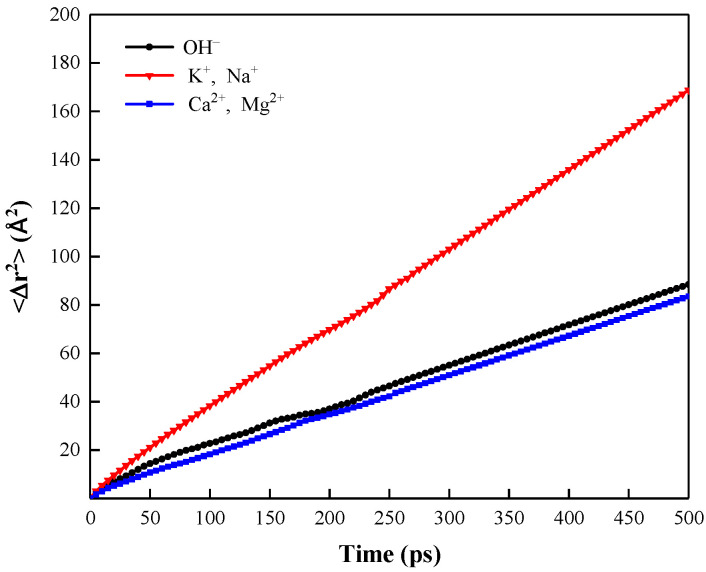
Mean square displacement of OH^−^ and cations in “SDBS/HPAM/OH^−^” system.

**Figure 12 polymers-15-00584-f012:**
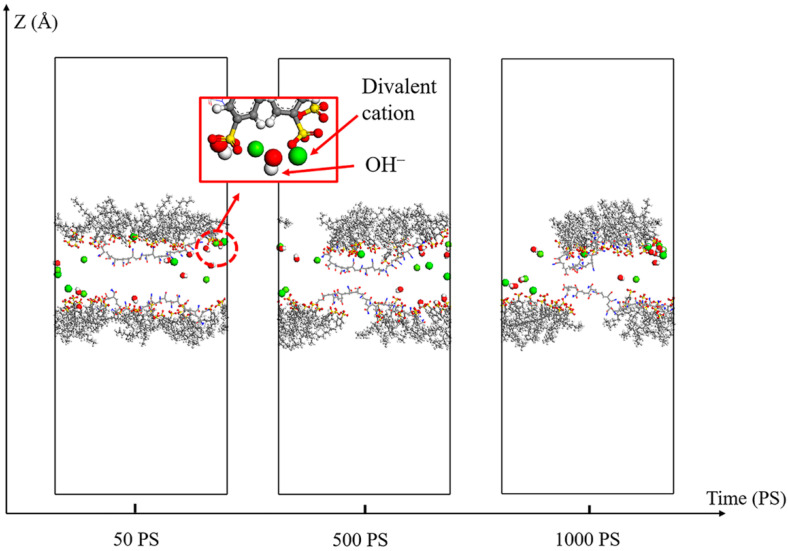
Snapshots of the liquid film in the “SDBS/HPAM/OH^−^” system during the dynamic relaxation process.

**Figure 13 polymers-15-00584-f013:**
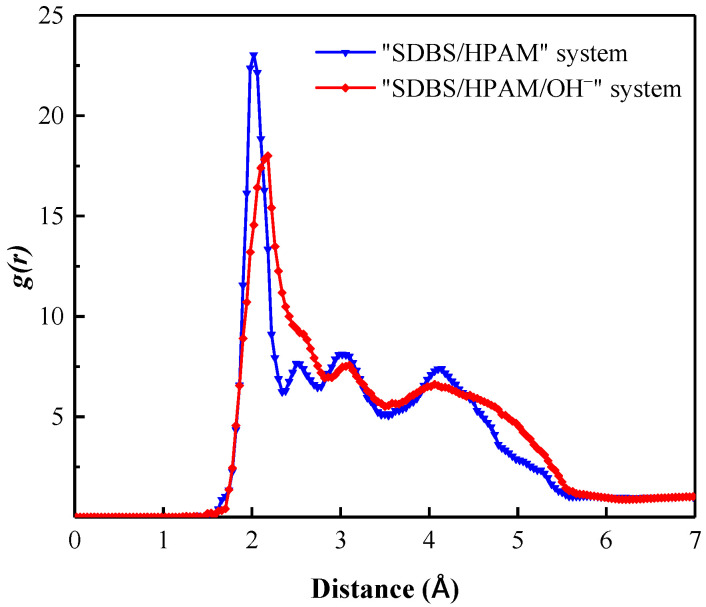
Radial distribution functions of -COO^−^ of HPAM molecules and surrounding cations in different foam systems.

**Figure 14 polymers-15-00584-f014:**
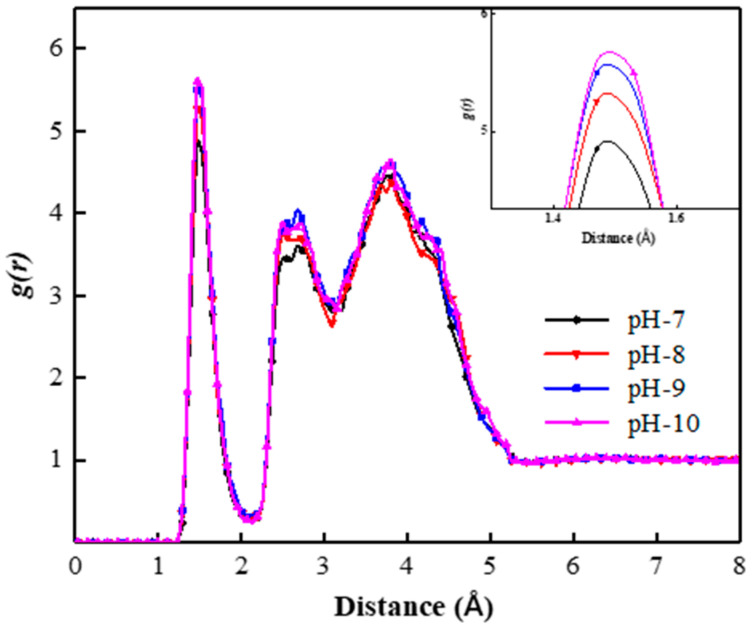
Radial distribution functions of SDBS head group and surrounding water molecules in foam systems with different pH.

**Figure 15 polymers-15-00584-f015:**
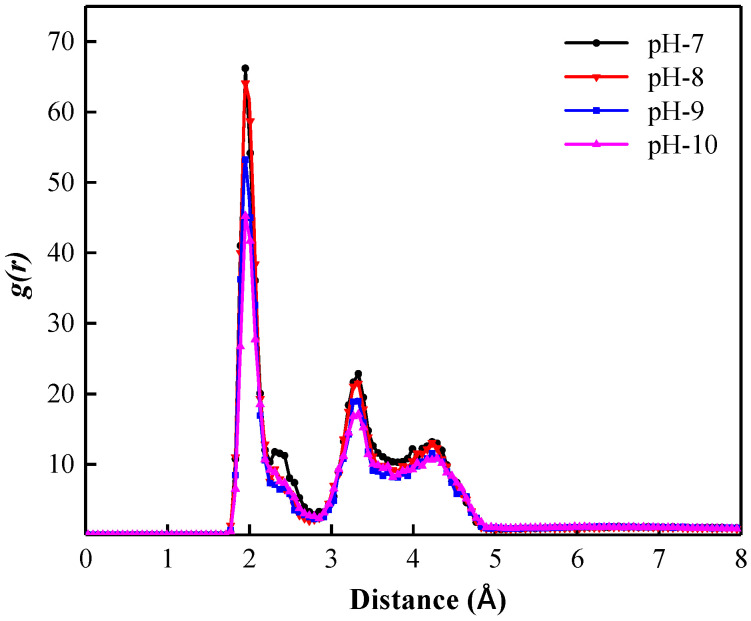
Radial distribution functions of SDBS head group and surrounding monovalent cations in foam systems with different pH.

**Figure 16 polymers-15-00584-f016:**
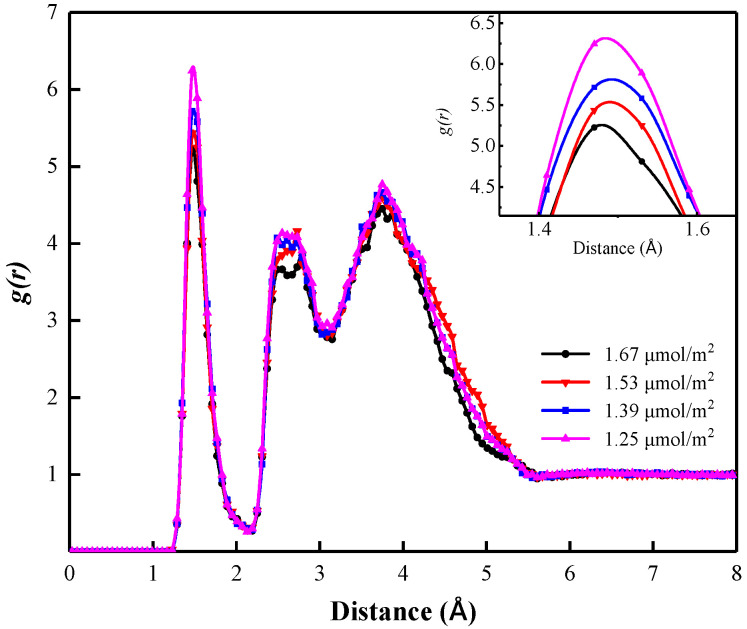
Radial distribution functions of SDBS head group and surrounding water molecules in foam systems with different SDBS concentrations.

**Figure 17 polymers-15-00584-f017:**
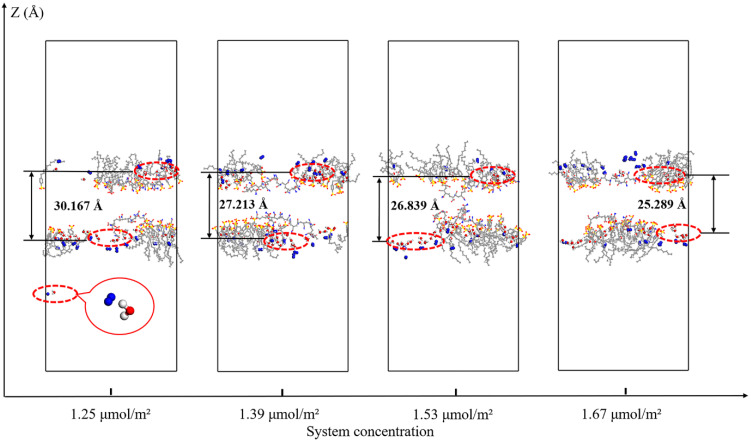
Snapshots of the gas–liquid interface in foam systems with different SDBS concentrations after dynamic relaxation equilibrium.

**Figure 18 polymers-15-00584-f018:**
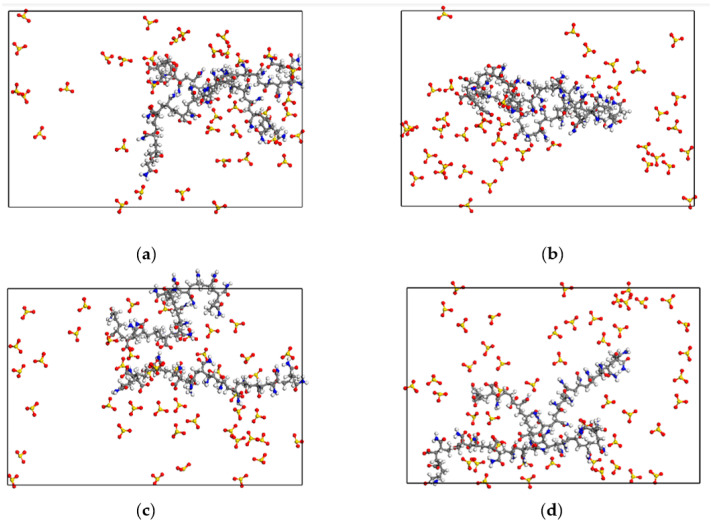
Sectional snapshots of the gas–liquid interface in foam systems with different SDBS concentrations after dynamic relaxation equilibrium. (**a**) 1.25 μmol/m^2^ system; (**b**) 1.39 μmol/m^2^ system; (**c**) 1.53 μmol/m^2^ system; (**d**) 1.67 μmol/m^2^ system.

**Figure 19 polymers-15-00584-f019:**
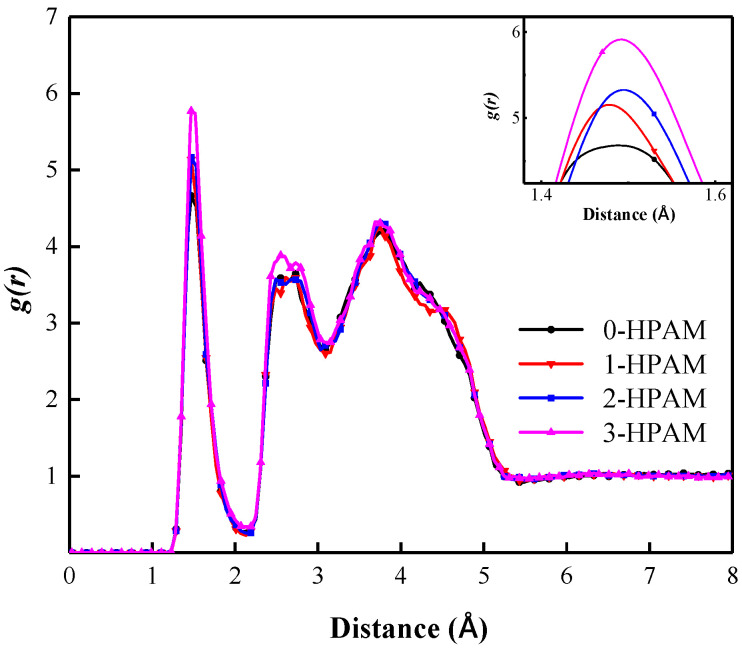
Radial distribution functions of SDBS head groups and surrounding water molecules in foam systems with different HPAM concentrations.

**Figure 20 polymers-15-00584-f020:**
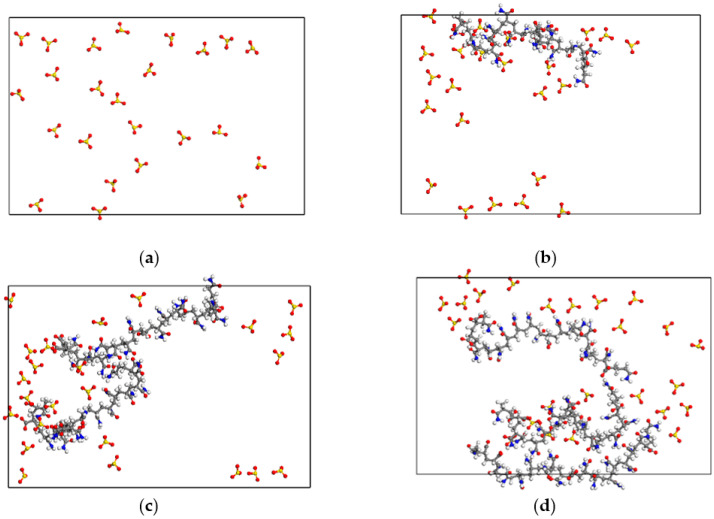
Sectional snapshots of the gas-liquid interface in foam systems with different HPAM concentrations after relaxation equilibrium. (**a**) “0-HPAM” system; (**b**) “1-HPAM” system; (**c**) “2-HPAM” system; (**d**) “4-HPAM” system.

**Figure 21 polymers-15-00584-f021:**
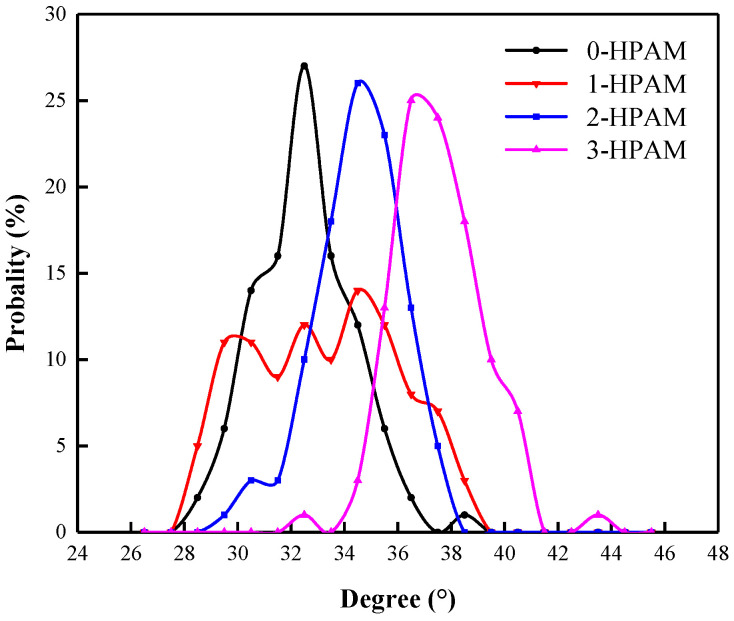
Distribution of SDBS molecular tail chain inclination angles in foam systems with different HPAM concentrations.

**Table 1 polymers-15-00584-t001:** Simulation scheme and molecular quantities in different foam systems.

Main Factor		Foam System	Molecular Quantities		
			SDBS	HPAM	OH^−^
Component composition		SDBS	48	-	-
		SDBS/HPAM	48	2	-
		SDBS/HPAM/OH^−^	48	2	10
Component properties	pH	pH 7	48	2	-
		pH 8	48	2	10
		pH 9	48	2	30
		pH 10	48	2	60
	SDBS concentration	1.25 μmol/m^2^	36	2	10
		1.39 μmol/m^2^	40	2	10
		1.53 μmol/m^2^	44	2	10
		1.67 μmol/m^2^	48	2	10
	HPAM concentration	0-HPAM	48	-	10
		1-HPAM	48	1	10
		2-HPAM	48	2	10
		3-HPAM	48	3	10

**Table 2 polymers-15-00584-t002:** Diffusion coefficients of water molecules in the hydration shell of the liquid film in different foam systems.

Foam System	*D*, 1 × 10^−5^ cm^2^·s^−1^
SDBS	2.06
SDBS/HPAM	1.94
SDBS/HPAM/OH^-^	1.86

**Table 3 polymers-15-00584-t003:** Coordination numbers of water molecules in the hydration shells of foam systems with different pH.

Foam System	Coordination Numbers of Water Molecule
pH 7	1.61
pH 8	1.74
pH 9	1.80
pH 10	1.82

**Table 4 polymers-15-00584-t004:** Coordination numbers of water molecules in the hydration shells of foam systems with different SDBS concentrations.

Foam System	Coordination Numbers of Water Molecule
1.67 μmol/m^2^	1.74
1.53 μmol/m^2^	1.79
1.39 μmol/m^2^	1.93
1.25 μmol/m^2^	1.98

**Table 5 polymers-15-00584-t005:** Coordination numbers of water molecules in the hydration shells of foam systems with different HPAM concentrations.

Foam System	Coordination Numbers of Water Molecule
0-HPAM	1.60
1-HPAM	1.63
2-HPAM	1.69
3-HPAM	1.86

**Table 6 polymers-15-00584-t006:** Gyration radius of HPAM molecules in foam systems with different HPAM concentrations.

Foam System	Gyration Radius, Å
1-HPAM	11.92
2-HPAM	11.45
3-HPAM	11.37

## Data Availability

Data are available from the corresponding author by request.

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
