# Peer review of "Molecular Dynamics Simulation of the Synergistic Effect of Alkali/Surfactant/Polymer on the Formation and Stabilization of Water-Based Foam Systems"

_polymers, 2023, doi:10.3390/polym15030584_

Round 1
Reviewer 1 Report (Previous Reviewer 1)
It seems that authors have made an effort to improve the manuscript, which can be acknowledged.
How do the authors explain the fact that g(r) does not reach asymptotically unity at large r????
Author Response
Attached please find the response in the attachment.

Reviewer 2 Report (Previous Reviewer 2)
This is my second review for the manuscript entitled: " Molecular dynamics simulation of the synergistic effect of alkali/surfactant/polymer on the formation and stabilization of water-based foam systems" by Yong Wang, Xinpeng Le, Xingwang Wang, Wenbo Liu, and Zhihua Wang (Manuscript ID: polymers-2068634)
The authors need to read the manuscript and correct any grammatical errors again. For example, in your addition to your manuscript to answering my major comment Q5, y” a large number of N2 molecules are diffused in the” should be “a large number of N2 molecules diffuse in the”.
Below find my follow-up questions/comments
Minor Comment Q4: Taking the absolute value DOES NOT MEAN that the units vanish, only the sign. Please include the units. 1 kJ/mole is certainly not the same as TJ/mole
Major comments
Q2: My question regarding the simulation cell size has to do with the accuracy of the simulation results. Having a smaller system would lead to inaccurate results. As such, increasing the size is not a mere arbitrary selection but has to be large enough so as not to have system-size effects, but yet small enough to be computationally tractable. I, therefore, ask again, have you doubled your simulation cell and compared your results with the smaller system (i.e., the one depicted in Fig 1). This should be done for at least one of your simulated systems.
Q4: The use of log-log plot is customary since it allows finding the power-law behavior in the two limits: small and large times. According to the Rouse model (see Doi & Edwards, The Theory of Polymer Dynamics, Ch. 4) at small times the MSD should scale with t^0.5 whereas at larger times, t>>tauR where tauR is the Rouse relaxation time, it follows the usual Fickian diffusion. Do you note any of these scalings or are your polymers too short?
Author Response
Attached please find the response in the attachment.
Thank you for your comments and time.

Reviewer 3 Report (Previous Reviewer 3)
While most of remarks and suggestions have been taken into consideration, there remain several points to improve and correct. Please revised again the manuscript along the list of indications (see report attached).

Author Response
Attached please find the response in the attachment.
Thank you for your comments and time.

Round 2
Reviewer 2 Report (Previous Reviewer 2)
This is my third review of the manuscript entitled: "Molecular dynamics simulation of the synergistic effect of alkali/surfactant/polymer on the formation and stabilization of water-based foam systems" by Yong Wang, Xinpeng Le, Xingwang Wang, Wenbo Liu, and Zhihua Wang (Manuscript ID: polymers-2068634).
Regarding your response to whether you have thoroughly checked whether you have system-size effects: You say you have undertaken all the necessary tests, but this is *not mentioned* in your revised manuscript. Please add this most important detail, which renders your results trustworthy.
Regarding your response to my request to have log-log plots of the MSD, I apologize for not realizing that you have been studying water molecules here. Obviously, in this case, you would have a Fickian behavior from t=0; therefore, there is no need to have a log-log plot.
Author Response
Thank you for your comments. Please find the author's response in the attachment.

Reviewer 3 Report (Previous Reviewer 3)
See attached

Author Response
Special thanks for your comments. Please find the author's response in the attachment.

This manuscript is a resubmission of an earlier submission. The following is a list of the peer review reports and author responses from that submission.
Round 1
Reviewer 1 Report
The manuscript cannot be accepted, since properties are wrongly reported. It seems that the authors are not in a position to judge whether their plots, and hence their simulations are correct. Moreover, while there is a good effort on the writing side, the overall presentation is rather poor with a lack of focus. I do not see how this manuscript can be improved in a revised version!
Reviewer 2 Report
The manuscript entitled " Molecular dynamics simulation of the synergistic effect of alkali/surfactant/polymer on the formation and stabilization of water-based foam systems" by Yong Wang, Xinpeng Le, Xingwang Wang, Wenbo Liu, and Zhihua Wang (Manuscript ID: polymers-2068634) presents molecular dynamics simulations of a water-based foam simulation system.
The authors should read the manuscript anew and correct the many grammatical errors. Please take the time to correct all of them after a close inspection of your manuscript.
Minor Comments
1) Sec. 2.1 “polymerization.33”. I guess that this should be a reference. Please correct. Same comment a few lines below for 34.
2) You say that in Fig. 4 you depict the g(r) of gas molecules without specifying the gas. Ince you only consider pure nitrogen for your simulations, you should explicitly state that throughout your manuscript.
3) It would be best if one were to use voids instead of holes when referring to the empty spaces seen in Fig. 7. Same comment for the use of the word holes in other parts of the manuscript.
4) Paragraph below Fig. 7. Please wherever you mention values of energy provide the units, which from Fig. 8 I assume are kJ/mole)
5) Fig. 11: I guess that a number of molecules, say water molecules, have been deleted here for clarity. You should mention this.
6) Fig. 16 should be bigger, as even after a considerable zoom-in I am unable to see clearly the snapshots shown.
7) Please define the radius of gyration used in Table 6, as I guess you are considering it as sqrt(<Rg^2>_eq)
Major Comments
1) p. 3: Regarding your initial configuration. You have already placed the SDBS molecules vertically in contact with the interface. Have you tried having them separately, say some distance from the interface, and allowing MD to bring them to their most favorable state, i.e., at the interface?
2) Fig. 1: From the figure, I see a rather small system. Have you checked that you do not have any system size effects? This is a very crucial test to be made as your systems are quite small.
3) Why have you used Compass as your preferred FF? Do you have any evidence of its validity for the systems you are studying? Reading your previous two papers, I was unable to find any such evidence. As you know, there are lots of FF out there and one should use the one more suitable to his/her particular case, i.e., which when used provides predictions closest to some available experimental data.
4) It would be best to have Fig. 3 as a log-log plot. Can you see different power-law regimes? Do you see similar dynamics at short times? Also, it would be best if you were to provide the time period which you considered for obtaining the diffusion coefficients of Table 2. Same for Fig. 10.
5) Fig. 4 requires further discussion. To me it seems that nitrogen molecules can penetrate up to 2 A away from the water-SDBS interface. What percent of the length, as projected to the z axis see your Fig. 5, of the SDBS is this? Also, you should specify the length of a SDBS molecule since one would eventually expect after this length that the g(r) of N2 molecules should reach unity. Finally, the term “aggregation of gas molecules” seems odd. Nitrogen molecules cannot aggregate with water as they do not reach the interface, or am I missing something?
6) Regarding Fig. 6, it would be interesting to see whether you also have, in addition to a tilt, a twist and a precession, see J. Phys. Chem. B 2008, 112, 1198-1211.
7) Regarding Fig. 7, I actually expect, and possibly even see in Fig. 7b, that water molecules form H bonds with the HPAM molecules. However, nothing is mentioned or discussed. Please do take the time to discuss this most important issue.
8) p. 15: “The water molecules at the gas−liquid interface have a tendency to move away from the liquid phase to the gas phase, and then overflow the liquid film.”. It would be much more clear if a series of snapshots were shown showing this. If this is what is shown in Fig. 17, I cannot see the SDMS-water interface which makes it hard to digest your points. Also, how is this controlled by the tilting of the SDMS chains?
Reviewer 3 Report
The manuscript “Polymers-2068634” reports a complete computational study of the properties of foams formed in three different compounds, including (1) sodium dodecylbenzene sulfonate (SDBS) foam system; (2) hydrolysed polyacrylamide (HPAM) and (3) an ionised SDBS/HPAM/OH- water-based foam. The authors investigated the microscopical mechanisms intervening in the formation and stability of foams including the three components listed above.
The methods reported are scientifically correct and also described with enough details. As I’ll indicate below, the water model adopted by the authors is not very appropriated, although the composition of the systems is quite realistic. There are major problems withFigures 2, 3 and 4, whereas the results reported in section 3.2 are much better computed and described, except for Fig. 12. The main problem is the lack of correct normalisation of the radial distribution functions.
In summary, the work reports interesting findings and it should be re-considered for publication in “Polymers”, but only after major revision of reported properties is performed. My strong advice is that current version should be rejected. In order to improve the manuscript, several remarks should be considered by the authors:
Major problems to address:
1. Water model SPC is one of worst available in the literature (see for instance Vega, Carlos, et al. "What ice can teach us about water interactions: a critical comparison of the performance of different water models." Faraday discussions 141, 251, 2009) and, given its importance in the present systems, the reasons for its use should be clearly explained or changed to, for instance, TIP4P-2005 (Abascal, Jose LF, and Carlos Vega. "A general purpose model for the condensed phases of water: TIP4P/2005." The Journal of chemical physics 123, 234505, 2005) or similar.
2. RDF in Figure 2 (page 5) are not normalised to 1 at long distances, please do it. In such a normalised version, I guess they will look essentially the same. So the role of water is system-independent, what should be addressed in the discussion. The current discussion in page 5 is not clear due to this drawback in the calculation of RDF.
3. In Figure 3, it is not clear that the length of MSD are long enough. Note that there three D’s are essentially the same around 2x10^(-5) cm^2/s. For me, no clear conclusion can be extracted from this. MSDs should be reported, at least, up to 500 ps.
4. In the same fashion as Fig.2, the validity of Figure 4 is not clear. Changes in the RDF of a gas around 17% are not significant enough to draw conclusive physical information. I suggest to withdraw such figure or simply indicate that in the three systems the effect of vapour is qualitatively the same. Further, these particular RDF are too noisy (too low statistics).
5. Figs. 5-11 are much better and well reported, but again Figure 12 has the same problems as Figs. 2 and 4. Please normalise and improve the statistics in order to have meaningful results. Current results are not conclusive.
6. On Figures 13, 14, 15 and 18, proper normalisation is again missed in all them. Consequently, coordination numbers reported in Tables 3, 4 and 5 are not clearly meaningful and should be recalculated after normalised RDFs.
7. Conclusions (1) and (3) should be revised after new RDFs are reported.
Minor details:
1. Acronym “HPAM” undefined in page 2.
2. Reference to “Materials Studio 8.0” missed in page 4.
3. Page 6, Stocks-Einstein has to be Stokes-Einstein.
4. English spelling and style needs revision throughout the manuscript.